METHODS AND RESOURCES

# Dating genomic variants and shared ancestry in population-scale sequencing data

**Patrick K. Albers** [ID]***, Gil McVean** [ID]

Big Data Institute, Li Ka Shing Centre for Health Information and Discovery, University of Oxford, Oxford, United Kingdom

* patrick.albers@bdi.ox.ac.uk

**Data Availability Statement:** We make all data, including variant age estimation and pairwise ancestry results, as well as the movies generated

## Abstract

The origin and fate of new mutations within species is the fundamental process underlying evolution. However, while much attention has been focused on characterizing the presence, frequency, and phenotypic impact of genetic variation, the evolutionary histories of most variants are largely unexplored. We have developed a nonparametric approach for estimating the date of origin of genetic variants in large-scale sequencing data sets. The accuracy and robustness of the approach is demonstrated through simulation. Using data from two publicly available human genomic diversity resources, we estimated the age of more than 45 million single-nucleotide polymorphisms (SNPs) in the human genome and release the Atlas of Variant Age as a public online database. We characterize the relationship between variant age and frequency in different geographical regions and demonstrate the value of age information in interpreting variants of functional and selective importance. Finally, we use allele age estimates to power a rapid approach for inferring the ancestry shared between individual genomes and to quantify genealogical relationships at different points in the past, as well as to describe and explore the evolutionary history of modern human populations.

## Introduction

Each generation, a human genome acquires an average of about 70 single-nucleotide changes through mutation in the germline of its parents [1]. Yet while, at a global scale, many millions of new variants are generated each year, the vast majority are lost rapidly through genetic drift and purifying selection. Consequently, even though the majority of variants themselves are extremely rare, the majority of genetic differences between genomes result from variants found at global frequencies of 1% or more [2], which may have appeared thousands of generations ago. Genome sequencing studies [3] have catalogued the vast majority of common variation (estimated to be about 10 million variants [4]), and, at least within coding regions and particular ancestries, to date, more than 660 million variants genome-wide have been reported [5], many of them at extremely low frequency [2].

Despite the importance of genetic variation in influencing quantitative traits and risk for disease, as well as providing the raw material on which natural selection can act, relatively little

from our results, publicly available online: https://human.genome.dating

**Funding:** Funded by the Wellcome Trust (100956/Z/13/Z to GM, 099685/Z/12/Z to PKA) and the Li Ka Shing Foundation (to GM). The funders had no role in study design, data collection and analysis, decision to publish, or preparation of the manuscript.

**Competing interests:** I have read the journal's policy and the authors of this manuscript have the following competing interests: GM is a shareholder in and non-executive director of Genomics PLC, and is a partner in Peptide Groove LLP. PKA is a shareholder in and a director of BioMe Oxford Ltd.

**Abbreviations:** AUC, area under the curve; BMRC, Biomedical Research Computing; CCF, cumulative coalescent function; CIF, coalescent intensity function; EDAR, Ectodysplasin A Receptor gene; ETPI, equal-tailed probability interval; GEVA, Genealogical Estimation of Variant Age; HLA, Human Leukocyte Antigene; HMM, hidden Markov model; IBD, identity by descent; IPG, Illumina Platinum Genomes Project; LCT, Lactase gene; LOESS, locally estimated scatterplot smoothing; LP, lactase persistence; MCM6, Minichromosome Maintenance Complex Component 6 gene; MRCA, most recent common ancestor; NHS, National Health Service; NIHR, National Institute for Health Research; PCA, principal component analysis; PolyPhen-2, Polymorphism Phenotyping v2 software; PRDM9, PR domain zinc finger protein 9; PSMC, pairwise sequentially Markovian coalescent; RMSLE, root mean-square $\log_{10}$ error; SGDP, Simons Genome Diversity Project; SIFT, Sorting Intolerant From Tolerant software; SNP, single-nucleotide polymorphism; TGP, 1000 Genomes Project; TMRCA, time to the most recent common ancestor; ZEB1, Zinc finger E-box–binding homeobox 1 gene.

attention has been paid to inferring the evolutionary history of the variants themselves, with notable exceptions of evolutionary importance, particularly those affecting geographically varying traits such as skin pigmentation, diet, and immunity [6–8]. Rather, attention has focused on the indirect use of genetic variation to detect population structure [9, 10], identify related samples [11, 12], infer parameters of models of human demographic history [13, 14], and estimate the time of ancestral population divergence [15]. The evolutionary history of classes of variants contributing to polygenic adaptation (for example, those affecting height [6, 16, 17]) or causing potential loss of gene function [18] has received attention, though rarely at the level of specific variants. Previous work on rare variants has identified ancestral connections between individuals and populations [19–21] and demonstrated evidence for explosive population growth [22]. Nevertheless, to date, no comprehensive effort has been made to infer the age, place of origin, or pattern of spread for the vast majority of variants.

We have developed a method for estimating the age of genetic variants; that is, the time of origin of an allele through mutation at a single locus. Our approach, which we refer to as the Genealogical Estimation of Variant Age (GEVA), is similar to existing methods that involve coalescent modeling to infer the time to the most recent common ancestor (TMRCA) between individual genomes [13, 23, 24]. However, these methods typically operate on a discretized timescale [13], utilize only a fraction of the information available in larger sample data [25], or employ approximations to overcome computational complexity [14, 15, 26]. Alternate approaches, in particular those that have been used to indicate allele age, are often strongly parameterized and make assumptions about demographic history, selection, or the shape of genealogical trees [27–30]. Recently proposed approaches for genealogical inference are also able to estimate allele age by placing mutation events on the branches of reconstructed trees at variable sites [31, 32]. Our work is motivated by the desire for a fast and scalable nonparametric estimator of allele age that makes no assumptions about the demographic or selective processes that shaped the underlying genealogy and that is robust to the frequency and types of error found in modern whole-genome population sequencing studies. We learn about the age of a mutation by combining probabilistic distributions of the TMRCA between hundreds or thousands of genomes on a continuous timescale, irrespective of contemporary allelic distributions or historic population boundaries and without the need to fully reconstruct genealogies.

The approach used within GEVA is outlined in Fig 1. A copy of the piece of the ancestral chromosome on which the mutation occurred is still present today in the individuals carrying the derived allele (Fig 1A). Over time, additional mutations have accumulated along the inherited sequence (haplotype), and its length has been broken down by recombination during meiosis in each generation. We locate this ancestral segment (often referred to as an identity-by-descent [IBD] region) relative to the position of a given target variant using a hidden Markov model (HMM) constructed empirically from sequencing data to provide robustness to realistic rates of sequencing and genotyping error (Fig 1B). By measuring the impact of mutation and recombination on the segments shared between pairs of haplotypes, we infer the TMRCA using probabilistic models to accommodate the stochastic nature of mutation and recombination processes (Fig 1C). Moreover, we make full use of the information available in whole-genome sequencing data to perform comparisons between pairs of chromosomes that both carry the mutation (concordant pairs) and pairs in which one carries the mutation and the other carries the ancestral allele (discordant pairs), thereby considering genealogical relationships that are both younger and older, respectively, than the time of mutation. Information from hundreds or thousands of haplotype pairs is then combined within a composite-likelihood framework to obtain an approximate posterior distribution on the age of the derived allele (Fig 1D). One benefit of our method is that we can increase the number of pairwise TMRCA inferences incrementally to update age estimates or to combine information across

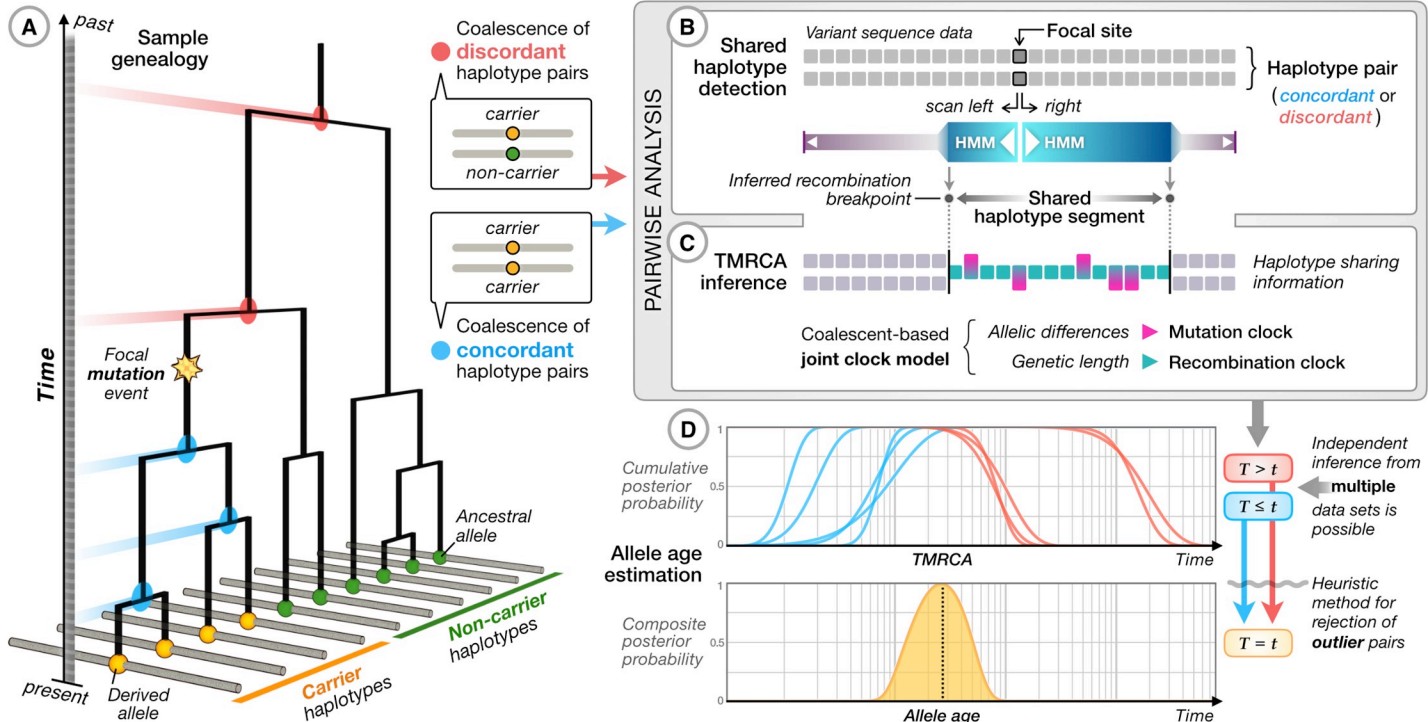

**Fig 1. Overview of the GEVA method.** (A) At the chromosomal location of a variant, there exists an underlying (and unknown) genealogical tree describing the relationship between the samples. We assume that the derived allele (inferred by comparison to outgroup sequences) arose once in the tree. For concordant pairs of carrier chromosomes (yellow terminal nodes), their MRCAs (blue nodes) occur more recently than the focal mutation event. For discordant pairs of chromosomes, between the ancestral allele (green terminal nodes) and the derived allele, the MRCAs (red nodes) are older than the focal mutation. (B) For each pair of chromosomes (concordant and discordant), we use a simple HMM with an empirically calibrated error model to estimate the region over which the MRCA does not change; that is, the distance to the first detectable recombination event either side of the focal position along the sequence. From the inferred ancestral segment, we obtain the genetic distance and the number of mutations that have occurred on the branches leading from the MRCA to the sample chromosomes. (C) For each pair of chromosomes, we use probabilistic models (see S1 Text) to estimate the posterior distribution of the TMRCA, represented as cumulative distributions of having coalesced for concordant pairs (blue) and of having not coalesced for discordant pairs (red). (D) An estimate of the composite posterior distribution for the time of origin of the mutation is obtained by combining the cumulative distributions for concordant and discordant pairs. Informally, the mutation is expected to be older than concordant and younger than discordant pairs. In practice, this composite-likelihood–based approach results in approximate posteriors that are overconfident; hence, they are summarized by the mode of the distribution. Additional filtering steps are carried out to remove inconsistent pairs of samples (see S1 Text). GEVA, Genealogical Estimation of Variant Age; HMM, hidden Markov model; MRCA, most recent common ancestor; TMRCA, time to the most recent common ancestor.

many data sources to improve the genealogical resolution from a wider distribution of independently sampled genomes. We additionally use a heuristic method for rejecting outlier pairs to improve robustness to low rates of data error and recurrent mutation. Full details are given in S1 Text.

## Results

### Simulation study

To validate GEVA, we performed coalescent simulations under different demographic models; see Fig 2. Using a standard coalescent model with constant mutation and recombination rates, we found low bias (relative error, $\varepsilon = 0.268$; see S2 Text) for allele age estimates and high correlation between true and inferred age (Spearman's $\rho = 0.953$; Fig 2A). We compared our approach for estimating the TMRCA to the computationally more demanding pairwise sequentially Markovian coalescent (PSMC) methodology [13], which forms the basis of many applications in ancestral inference [14, 26]. PSMC estimates a model of the demographic history between pairs of chromosomes (over a discretized grid of time intervals) and can, for

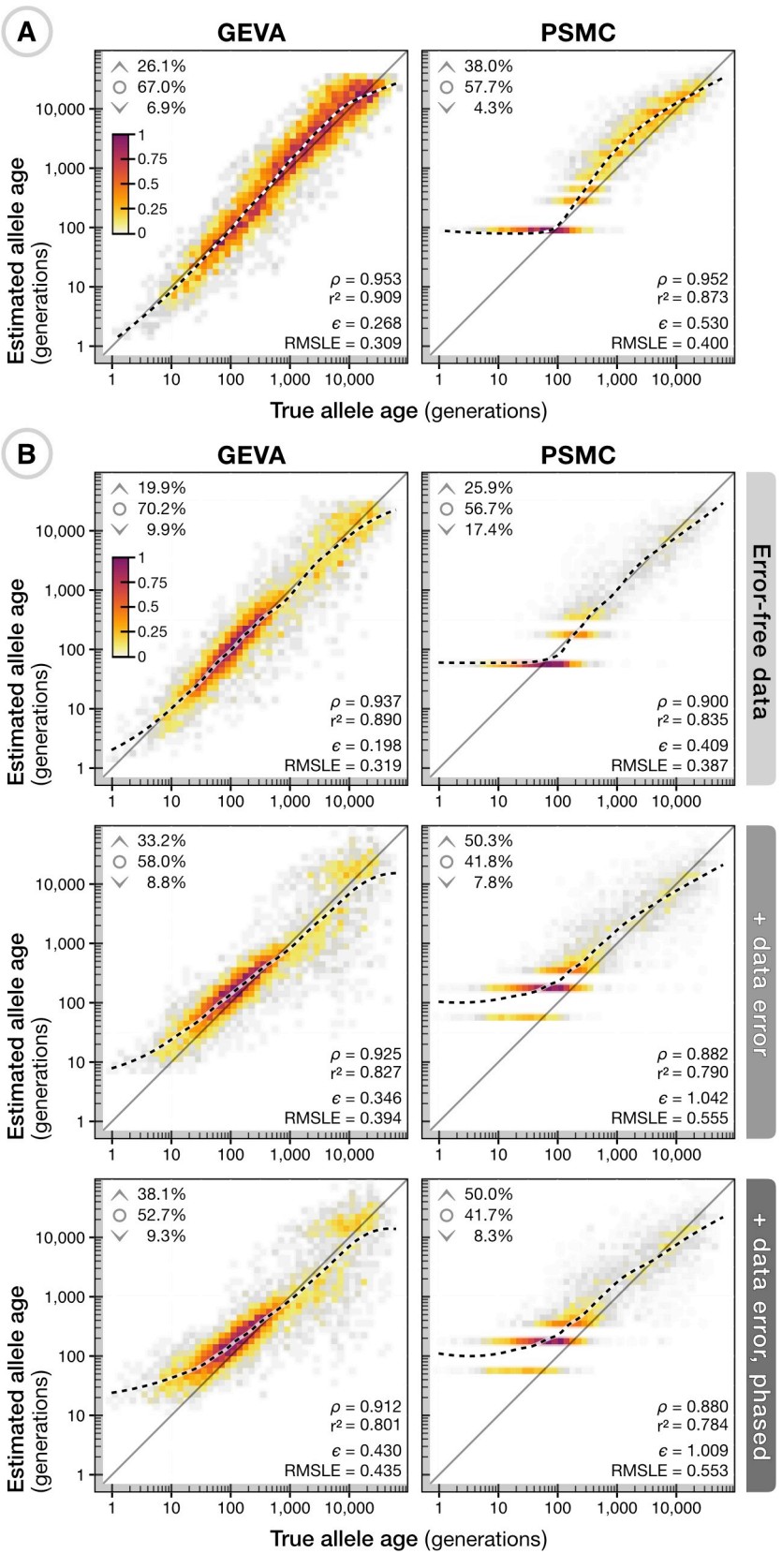

**Fig 2. Validation of GEVA through coalescent simulations.** (A) Density scatterplots showing the relationship between true allele age (geometric mean of lower and upper age of the branch on which a mutation occurred; $x$ axis) and estimated allele age ($y$ axis), using GEVA with the in-built HMM methodology (left) and PSMC (right) for the same set of 5,000 variants. Data were simulated under a neutral coalescent model with sample size $N = 1,000$, effective population size $N_e = 10,000$, and with constant and equal rates of mutation ($\mu = 1 \times 10^{-8}$) and recombination ($r = 1 \times 10^{-8}$) per site per generation. Variants were sampled uniformly from a 100-Mb chromosome, with allele count $1 < x < N$. Colors indicate relative density (scaled by the maximum per panel). Upper inserts indicate the fraction of sites where the point estimate (mode of the composite posterior distribution) of allele age lies above the upper age of the branch on which it occurred ($\wedge$), below the lower age ($\vee$), or within the age range of the branch ($\circ$). Lower inserts indicate the Spearman rank correlation statistic $\rho$, squared Pearson correlation coefficient (on log scale) $r^2$, interval-adjusted bias metric (see S2 Text) $\varepsilon$, and RMSLE. Also shown is an LOESS fit (second-degree polynomials, neighborhood proportion $\alpha = 0.25$; dashed line). (B) The relationship between true and inferred ages for 5,000 variants sampled uniformly from a simulation under a complex demographic model with $N = 1,000$, $N_e = 7,300$, $\mu = 2.35 \times 10^{-8}$, and variable recombination rates from human chromosome 20 (63 Mb). Allele age was estimated on haplotype data as simulated and without error (top), with error generated from empirical estimates of sequencing errors (middle), and with additional error arising from in silico haplotype phasing; see S2 Text. Allele age was estimated using scaling parameters as specified for each simulation. A further breakdown of results using mutation and recombination clocks alone, as well as the inferred pairwise TMRCAs, is available for A (S1 Fig) and B (S2 Fig, S3 Fig, S4 Fig). GEVA, Genealogical Estimation of Variant Age; HMM, hidden Markov model; LOESS, locally estimated scatterplot smoothing; PSMC, pairwise sequentially Markovian coalescent; RMSLE, root mean-square $\log_{10}$ error; TMRCA, time to the most recent common ancestor.

every position in the genome, return the inferred posterior distribution on the TMRCA, thus enabling a composite-likelihood estimator of allele age as in GEVA.

We found that PSMC-based age estimations performed similarly well to GEVA ($\rho = 0.952$), though the time discretization increased bias ($\varepsilon = 0.530$) and, in particular, led to overestimation of the age for the youngest variants. We note that PSMC was not designed strictly for this purpose and hence is not optimized for estimating allele age. Conversely, pairwise estimates of the TMRCA between concordant haplotypes were highly correlated with true TMRCA in both GEVA ($\rho = 0.922$) and PSMC ($\rho = 0.919$), but the correlation for discordant pairs was lower in GEVA ($\rho = 0.586$) compared to PSMC ($\rho = 0.766$); see S1 Fig. Such differences in relation to estimating allele age with high accuracy are tolerated because the time of mutation is estimated from the composite distribution of TMRCA posteriors from many pairwise comparisons performed at a single locus. Events that occurred distant in time (relative to the time of mutation) will have little influence on the estimate.

Under a complex demographic model that recapitulates the human expansion out of Africa and with empirical and variable recombination rates, age estimation in GEVA maintained a similarly high level of accuracy ($\varepsilon = 0.198$, $\rho = 0.937$; Fig 2B). In this situation, although PSMC modeled the more dynamic demographic histories between haplotypes with higher accuracy (correlation of true and inferred TMRCA for discordant pairs: $\rho = 0.915$) compared to GEVA ($\rho = 0.775$), the time discretization resulted in artifacts at more recent times (TMRCA for concordant pairs: $\rho = 0.892$) that were not present in GEVA ($\rho = 0.932$), leading to worse performance when estimating the age ($\varepsilon = 0.409$, $\rho = 0.900$; S2 Fig), with the addition of substantial computational cost. We next introduced realistic data complications by reproducing empirically estimated genotype errors in simulated data (see S2 Text), as well as errors arising through in silico haplotype phasing (Fig 2B). We found age estimation in GEVA to remain largely unbiased and strongly correlated with true age after the inclusion of data error ($\varepsilon = 0.346$, $\rho = 0.925$; S3 Fig) and after phasing ($\varepsilon = 0.430$, $\rho = 0.921$; S4 Fig). The PSMC-based approach continued to show higher bias and reduced correlation at the same set of variants, both after error ($\varepsilon = 1.042$, $\rho = 0.882$) and after phasing ($\varepsilon = 1.009$, $\rho = 0.880$). Reduced data quality resulting from sequencing errors may introduce false signals of pairwise differences seen between haplotypes, and phasing errors may lead to an underestimation of haplotype lengths at variants that are relatively young, for which we overestimate TMRCA and hence allele age (particularly for alleles younger than approximately 100 generations).

## Age of selected variants

To evaluate the performance of GEVA on empirical data, we first considered variants affecting the well-studied lactase persistence (LP) trait, for which numerous approaches, including the use of archaeological data, genetic data, and a biological understanding of the functional and evolutionary impact of previously associated variants, have resulted in consensus expectations for the age. The *LCT* gene encodes the lactase enzyme but is regulated by variants in an intron of the neighboring *MCM6* gene (Minichromosome Maintenance Complex Component 6). We estimated the age of the derived T allele of the rs182549 variant (G/A-22018), which is at a frequency of approximately 50% in European populations and forms part of a haplotype associated with LP [33]. Under a model that jointly considers mutational and recombinational information, we estimated the allele to be 688 generations old (Fig 3A), originating approximately 14,000 to 21,000 years ago, depending on assumptions about generation time in humans [34, 35]. Our estimate is based on data from two different sources, the 1000 Genomes Project (TGP) [2] and the Simons Genome Diversity Project (SGDP) [36], which, when estimated separately, give very similar ages (692 and 687 generations, respectively). The full result data set for this variant is available online: https://human.genome.dating/snp/rs182549. We obtained a similar age estimate of 693 generations for the derived A allele of the rs4988235 (C/T-13910) variant (see https://human.genome.dating/snp/rs4988235), which is also strongly associated with LP and in near perfect association with rs182549, though we note that there is evidence for multiple origins of the variant [37]. Previous estimates of the age of these variants range between 2,200 and 21,000 years [38], putting our estimates on the higher end of this range. Multiple sources of information suggest that these variants only achieved high frequency in European populations within the last 10,000 years (approximately 400 generations) [39] and that LP alleles were rare until the advent of dairy farming in Europe [40]. Our results therefore suggest that the mutation conferring the strongly selected phenotype (estimated to have a selection coefficient of up to 15% in European and up to 19% in Scandinavian populations [39]) was present for hundreds of generations before its rapid sweep through the population.

We next considered the protein-coding missense variant rs3827760 in the Ectodysplasin A Receptor (*EDAR*) gene, where the derived G allele (Val370Ala substitution) is found at high frequency in East Asian populations (87% in the TGP, 82% in the SGDP) and American populations (39% and 80%, respectively) and is associated with sweat, facial and body morphology, and hair phenotypes [41, 42]. We estimated the variant to be 1,456 generations old, approximately 29,000 to 44,000 years (Fig 3B; https://human.genome.dating/snp/rs3827760), again with strong concordance between the TGP (1,513 generations) and the SGDP (1,346 generations). Our estimate is consistent with previous estimates and limited evidence from ancient DNA studies [7, 44]. Our results further show that most individuals carrying the allele share a common ancestor close to the time the allele arose through mutation. This may suggest that the variant rapidly rose in frequency following its origin, which is consistent with previous findings of strong positive selection of this variant in East Asia [41]. We compared this result to variants of similar age (1,450 ± 100 generations) in the Atlas of Variant Age (see further below): we found 1,043,376 variants dated in the TGP, of which 2,073 have reached a frequency higher than 30% globally and only 130 above 80% in East Asian populations, demonstrating how unusual such a rapid rise in frequency is.

Finally, we considered the variant rs80194531, in which the derived allele causes an Asn78Thr substitution in the Zinc finger E-box–binding homeobox 1 (*ZEB1*) gene. The variant is reported as pathogenic for corneal dystrophy [43] but is present at 6% in African ancestry samples within the TGP or the SGDP. We estimated the age of the variant to be 5,866

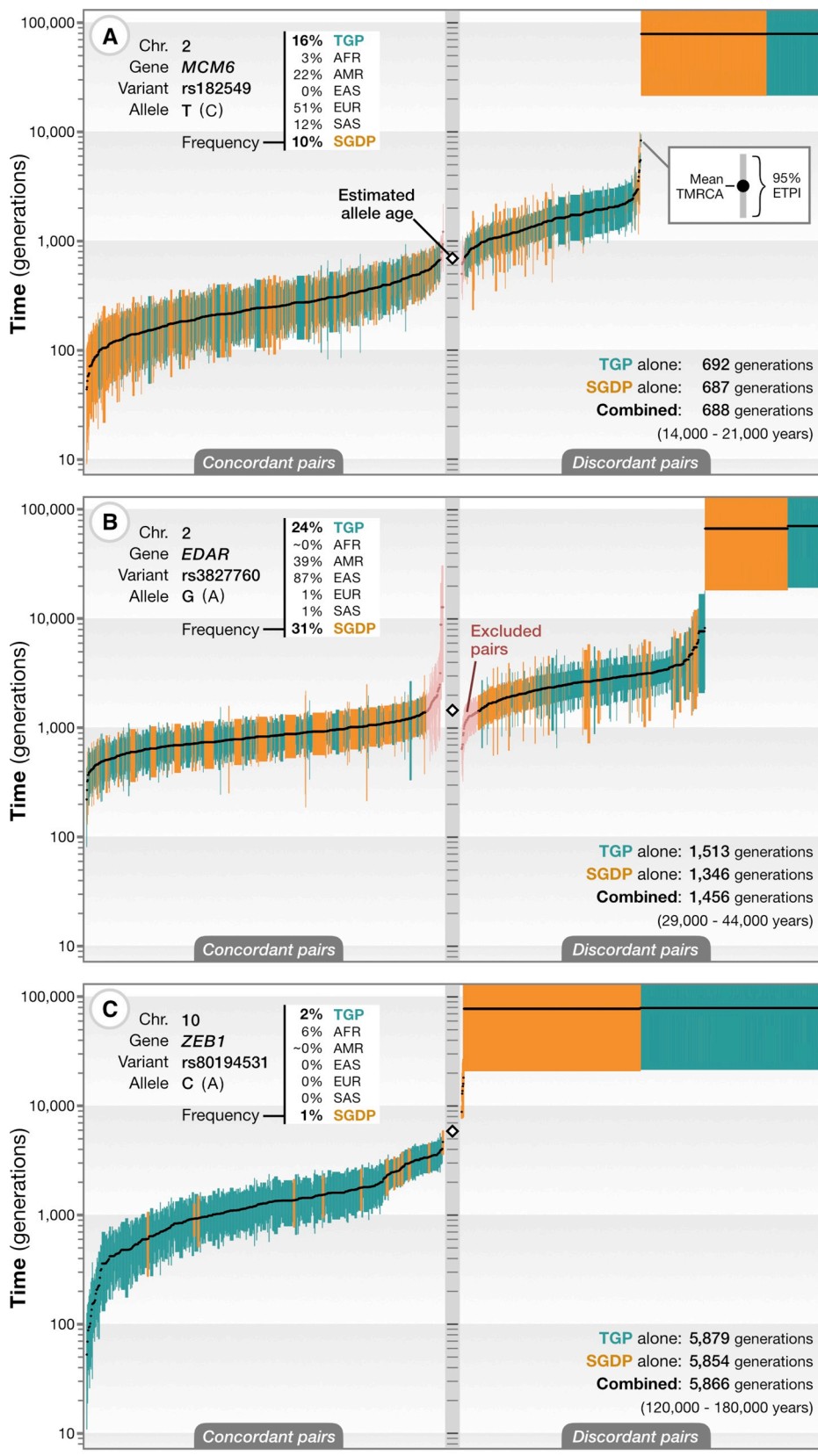

**Fig 3. Application of GEVA to 3 variants of phenotypic and selective importance.** (A) Estimated TMRCAs for concordant (left) and discordant (right) pairs of chromosomes for the derived T allele at rs182549, which lies within an intron of *MCM6* and affects regulation of *LCT* [33], which encodes lactase. Each bar reflects the approximate 95% credible interval (ETPI) for a pair, ordered by posterior mean (black dots). Data from the TGP (green) [2] and the SGDP (orange) [36] were used. The frequency of the variant in the SGDP, the TGP, and the different population groups in the TGP is shown (top left). The inferred allele age in generations from each data source and the combined estimate are shown (bottom right) and converted to an approximate age in years, assuming 20–30 years per generation. See https://human.genome.dating/snp/rs182549 for additional results. (B) As for panel A for the derived G allele of rs3827760, which encodes the Val370Ala variant in *EDAR* and is associated with sweat and facial and body morphology [41, 42]; also see https://human.genome.dating/snp/rs3827760. Our filtering approach is to remove the smallest number of concordant and discordant pairs necessary (shown in pink) to obtain concordant and discordant sets with nonoverlapping mean posterior TMRCAs. (C) As for panel A for the derived C allele of rs80194531, which encodes the Asn78Thr substitution in *ZEB1*, reported as pathogenic for corneal dystrophy [43]; also see https://human.genome.dating/snp/rs80194531. Abbreviations refer to ancestry groups. AFR, African; AMR, American; EAS, East Asian; *EDAR*, Ectodysplasin A Receptor gene; ETPI, equal-tailed probability interval; EUR, European; GEVA, Genealogical Estimation of Variant Age; *LCT*, Lactase gene; *MCM6*, Minichromosome Maintenance Complex Component 6 gene; SAS, South Asian; SGDP, Simons Genome Diversity Project; TGP, 1000 Genome Project; TMRCA, time to the most recent common ancestor; *ZEB1*, Zinc finger E-box–binding homeobox 1 gene.

generations old (120,000 to 180,000 years), again with consistency between the TGP and the SGDP (5,879 and 5,854 generations, respectively; Fig 3C; https://human.genome.dating/snp/rs80194531). Such an ancient age seems inconsistent with the reported dominant pathogenic effect [43]. Moreover, of the 1.2 million variants found at comparable frequencies (5%–7%) in African ancestry individuals within the TGP, we found that 46% were estimated to be younger than the rs80194531 allele, suggesting that this variant is in no way unusual.

## Distribution of allele age in the human genome

We next sought to characterize the age distribution of genetic variation across the human genome, for which we applied GEVA to more than 45 million variants identified in the TGP or the SGDP. More than 32 billion haplotype pairs were analyzed to estimate shared haplotype segments and TMRCAs. For variants present in both data sources (13.7 million), we additionally estimated the age by combining pairwise TMRCA distributions that were inferred independently in each sample after confirming that separately obtained age estimates agreed (Spearman's $\rho = 0.862$; see S5 Fig). We make this information, referred to as the Atlas of Variant Age for the human genome, publicly available as an online database (https://human.genome.dating). A breakdown by chromosome of the number of variants dated and haplotype pairs analyzed is given in S1 Table. Further details are given in S3 Text.

We find substantial variation in the relationship between estimated age and allele frequency, depending on the population in which frequency is measured and the geographical distribution of the variant (Fig 4A). Variants in African ancestry groups are typically older than in other groups and also have the greatest variance in age for a given frequency. For example, variants below 0.5% (within a given population) have a median age of 670 generations in African ancestry groups, 377 generations in East Asian ancestry groups, and 488 generations in Europeans (see S2 Table). The age distribution of variants restricted to a particular ancestry group (or shared between them) indicates the degree of connection between populations. For example, there are many variants up to 5,000 generations old (100,000 to 150,000 years) that are restricted to African ancestry groups yet are observed at frequencies up to 10%, but variants in this frequency range that are restricted to East Asian or South Asian ancestry groups are typically under 1,000 generations (20,000 to 30,000 years) or 1,300 generations old (26,000 to 39,000 years), respectively. Conversely, cosmopolitan variants that are shared among every ancestry group are typically older than 2,000 generations (40,000 to 60,000 years) despite being observed at global frequencies below 0.5% (S6 Fig). Variants restricted to

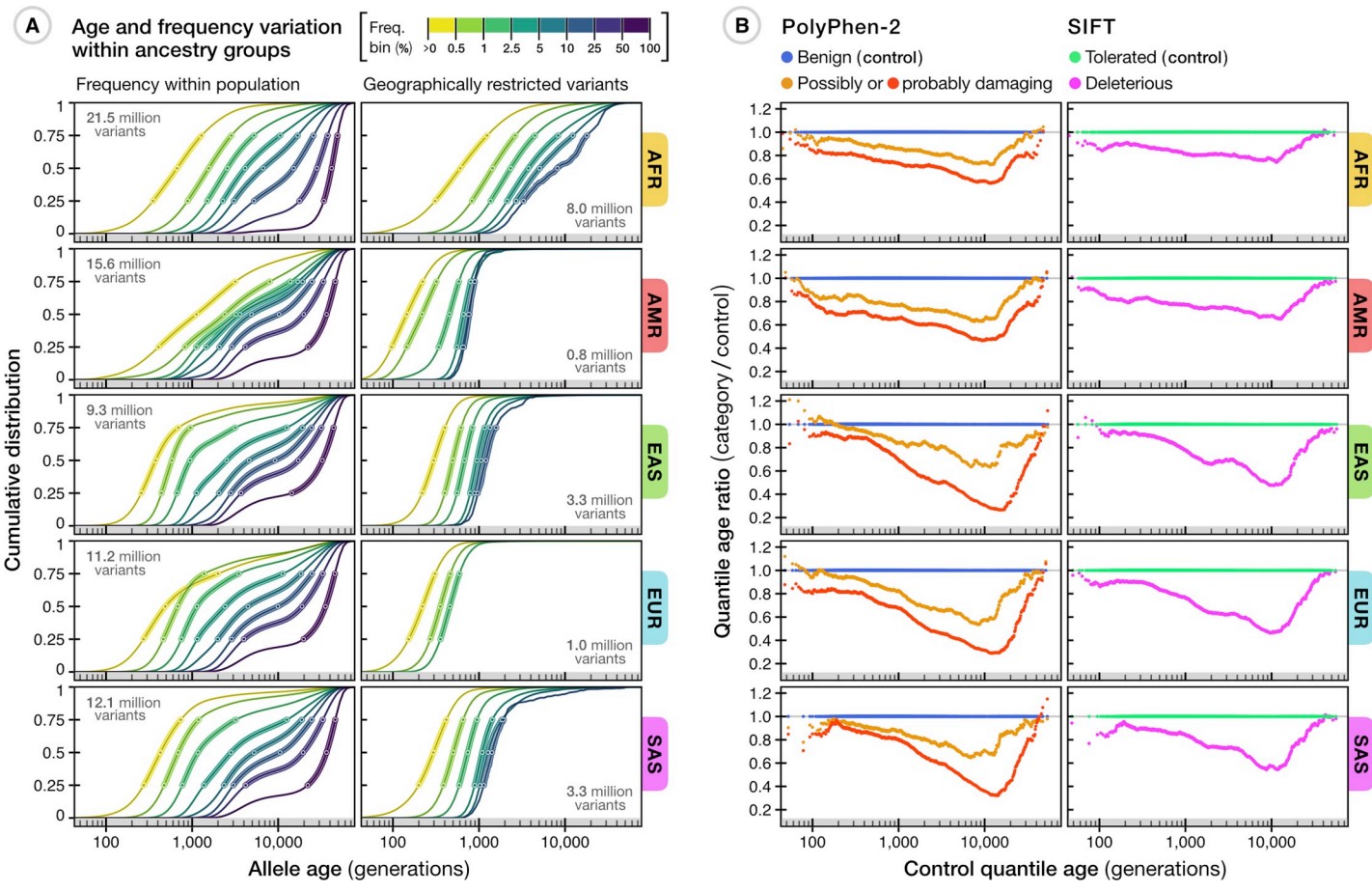

**Fig 4. Age distribution of variants among different human populations.** (A) The relationship between estimated allele age and frequency as observed within a given population group in the TGP sample. Of the 45.4 million variants available in the Atlas of Variant Age, 43.2 million were dated using TGP data alone; we excluded variants with low estimation quality and inconsistent ancestral allele information (see S3 Text), retaining 34.4 million variants. Each line shows the cumulative age distribution of variants within a given frequency bin (see legend) within a population group; circles indicate median and interquartile range. Panels on the left show the frequency-stratified cumulative distribution of estimated age for variants at nonzero frequencies as observed within a given ancestry group. The number of variants available per group is shown (top left). Panels on the right show the distributions of geographically restricted variants that only segregate within a group (number of available variants shown on bottom right). A summary of variants shared between different ancestry groups in the TGP is provided in S6 Fig. (B) Differences in allele age distributions for approximately 70,000 variants in the TGP that are annotated as impacting protein function by PolyPhen-2 (left) and SIFT (right), compared to a reference set of variants (those annotated as benign by PolyPhen-2 or tolerated by SIFT), matched for allele frequency within a given ancestry group. These results are presented in more detail in S7 Fig. AFR, African; AMR, American; EAS, East Asian; EUR, European; PolyPhen-2, Polymorphism Phenotyping v2 software; SAS, South Asian; SIFT, Sorting Intolerant From Tolerant software; TGP, 1000 Genomes Project.

American ancestry groups are typically younger than 750 generations (15,000 to 22,500 years), consistent with existing knowledge about the settlement of the Americas via the Bering land bridge that connected Asia and North America during the last glacial maximum around 15,000 to 23,000 years ago [45, 46]. We note, however, that recent admixture and the sampling strategies of the different data sets [47, 48] can have a strong impact on age distributions. For example, variants at high frequency within American populations but that are nevertheless restricted to just American and African populations are, on average, younger than lower-frequency variants (within American populations) with the same geographical restriction (S6 Fig). These variants likely arose recently within Africa and entered American populations through admixture, rising to high frequency through population bottlenecks [49]. Similarly, variants found only within European and African populations but that have a frequency below 0.5% in Europeans are, on average, older than variants observed at higher frequencies in

Europeans and older than variants restricted to only Europeans in the same frequency range, suggesting recent gene flow from Africa into Europe.

Such heterogeneity in the relationship between allele age and frequency, coupled with heterogeneous and unknown sampling strategies, complicates the use of frequency as a means of assessing variants for potential pathogenicity during the interpretation of individual genomes. The Atlas of Variant Age potentially offers a more direct approach for screening variants, given the high probability of elimination of nonrecessive deleterious variants within a few generations [50]. To assess the value of allele age in the interpretation of potentially pathogenic variants, we estimated the ages of variants that had effects predicted as damaging by Polymorphism Phenotyping v2 software (PolyPhen-2) [51] or deleterious by Sorting Intolerant From Tolerant software (SIFT) [52] in the TGP (Fig 4B). Of the approximately 70,000 variants analyzed, 50% of damaging and 49% of deleterious variants were estimated to have arisen within the last 500 generations (10,000 to 15,000 years), compared to 41% of benign (PolyPhen-2) and 42% of tolerated (SIFT) variants (S7 Fig). Compared with control sets of variants (those annotated as benign or tolerated and matched for allele frequency within the focal ancestry group), variants annotated as damaging or deleterious had a notable dearth of older variants (>1,000 generations) for a given frequency, consistent with theoretical expectations and previous findings [19, 53, 54]. Our results suggest that old alleles can largely be excluded from consideration of pathology (though recent origin is not evidence in favor of pathogenicity).

## Shared ancestry

Finally, we investigated the extent to which patterns of sharing of variants of different ages could power approaches for learning about genealogical history. Previous work has highlighted the descriptive value of genetic variants in identifying individuals who share recent common ancestry and patterns of demographic isolation and migration [10, 55, 56], though it has also highlighted the challenges of interpreting the output of approaches such as principal component analysis (PCA) [57, 58]. Conversely, numerous model-based approaches have been developed that use patterns of variant and haplotype sharing to infer underlying demographic parameters [14, 59–61], though these typically make strong simplifying assumptions about the space of possible histories.

Here, we present a nonparametric approach for combining descriptive and inferential approaches to learn about ancestral connections between individual genomes (and groups of individuals) based on variant age information. Unlike existing methods that assign time-invariant ancestry proportions to individual genomes by reference to contemporary populations [9, 62], we can estimate the fraction a given genome shares because of common ancestry with any other genome at different points in time, referred to as the cumulative coalescent function (CCF). We use a fast dynamic-programming approach to estimate a maximum likelihood CCF between any pair or group of individuals (Fig 5A; see S4 Text). Using simulated data, we found that inference of ancestral relationships between individuals (CCFs inferred from estimated allele ages) correctly reflected differences of relatedness among individuals within and between different ancestry groups and revealed patterns qualitatively consistent with past demographic events (see S5 Text), though we note that uncertainty in variant age estimates may cause oversmoothing of coalescent profiles, in particular in the distant past (>20,000 generations ago).

To illustrate the value of this nonparametric approach in describing the history of individuals and groups, we first considered the coalescent history between a single individual of American (Puerto Rican) ancestry from the TGP (ID: HG00733) and all others in the TGP sample, using GEVA age estimates for variants on chromosome 5 (Fig 5B). As a positive control, we

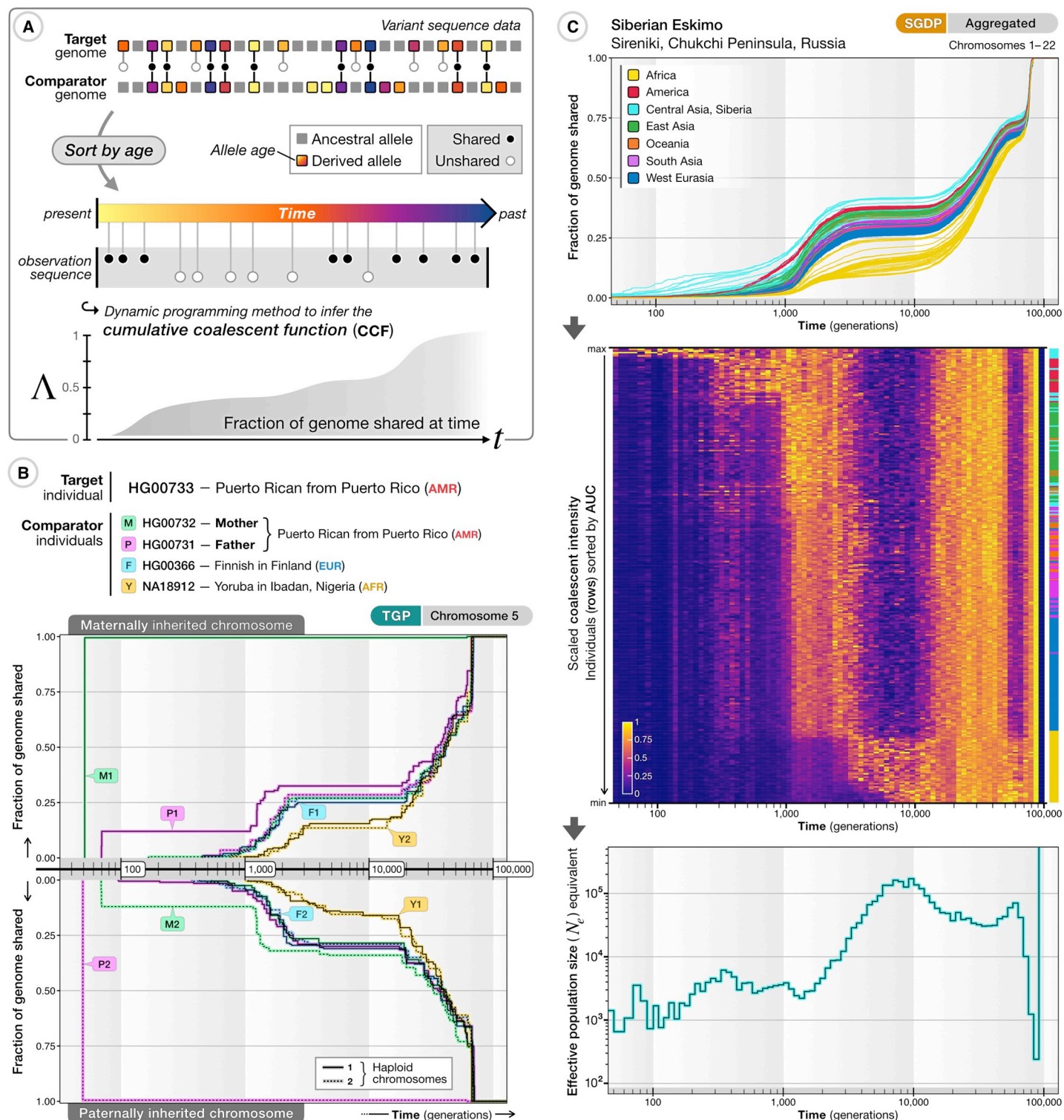

**Fig 5. Age-stratified variant sharing to characterize ancestral relatedness.** (A) Overview of approach for estimating the CCF for a pair of haplotypes, the fraction of the genomes of the two samples that have coalesced by a given time. Derived variants within a target genome are identified, their estimated ages are obtained from the Atlas of Variant Age, and their presence (black circles) or absence (white circles) in another (comparator) genome of interest is recorded. Variants are sorted by allele age (indicated by color), $t$, to obtain a naive maximum likelihood estimate of the CCF, $\Lambda(t)$, using dynamic programming (assuming independence of variants and ignoring error in variant age estimates). (B) Selected pairwise CCFs for the two haploid genomes (chromosome 5) of individual HG00733 (top: maternally derived; bottom: paternally derived) of a Puerto Rican individual from the TGP compared to 8 haplotypes from 4 individuals, including their mother and father. Maternal and

paternal genomes were used for phasing; hence, the inferred parental genomes are the transmitted (and untransmitted) genomes. The CCFs inferred with genomes from the entire TGP sample is shown in S1 Movie. The full result data set for HG00733 is available at https://human.genome.dating/ancestry/HG00733. (C) Inferred genome-wide CCFs (averaged per diploid individual across autosomes) for a Siberian Eskimo from the SGDP (ID: S_Eskimo_Sireniki-1) to all other sampled individuals (top panel). Colors indicate ancestry by geographic region (see legend). The CCF can also be expressed as a CIF (middle panel) to reflect the increase in shared ancestry within a given time period. Each row represents an individual from the SGDP, ordered by the AUC of the CCF and scaled such that the maximum per column is equal to one. The color bar (right) indicates the ancestry group of sorted individuals. The CIF within any time epoch can be expressed as an effective population size ($N_e$ equivalent) from the maximum over reference samples, providing a summary of the rate at which common ancestor events occurred (bottom panel). The full result data set for this individual is available at https://human.genome.dating/ancestry/S_Eskimo_Sireniki-1. AFR, African; AMR, American; AUC, area under the curve; CCF, cumulative coalescent function; CIF, coalescent intensity function; EUR, European; SGDP, Simons Genome Diversity Project; TGP, 1000 Genomes Project.

included the parents of HG00733 (HG00732 and HG00731), who reach a CCF of near 1 in the most recent epoch (though note that the parents were used for haplotype phasing, which estimates transmitted haplotypes; hence the CCF reaching 1 rather than the expected one-half). We show the ancestry of HG00733 shared with every individual in the TGP sample in S1 Movie; also see https://human.genome.dating/ancestry/HG00733. Within the first 100 generations, we see additional coalescence with the untransmitted parental chromosomes and other individuals from the Puerto Rican sample. The earliest common ancestry outside Puerto Rico is seen with a European individual in Spain (paternal side; approximately 60 generations ago) and a Mexican ancestry individual (maternal side; approximately 80 generations ago). Coalescence with individuals sampled from outside the Americas occurs further back in time (>100 generations ago), initially with European individuals (predominantly around 300 to 600 generations ago), then uniformly with non-African individuals around 1,500 to 4,000 generations ago, more strongly with African individuals around 5,000 to 15,000 generations ago, and uniformly with all individuals around 20,000 generations ago. Because of the impact of data errors on age estimation of recent variants (highlighted above), the absolute timings of the early events are likely substantially overestimated. However, we expect the relative ordering of events to be robust and consistent among sample comparisons (see S5 Text).

The CCFs to all other members of a reference panel (averaged across all chromosomes in both haploid genomes) provide an overview of the genealogical relationships for a target individual. As an example, we inferred the CCF profiles of a Siberian Eskimo to all other individuals in the SGDP (Fig 5C, top), showing common ancestry to other Central Asian and Siberian individuals within a few hundred generations, substantial common ancestry with American individuals before 1,000 generations, and typically more recent common ancestry with East Asians than West Eurasians or Africans; see https://human.genome.dating/ancestry/S_Eskimo_Sireniki-1. Notably, relatively little additional coalescence is seen during a period around 2,000 to 10,000 generations ago, which is a pattern shared among non-African individuals (also see S1 Movie) and agrees with previous findings of a period of reduced coalescence, peaking 100,000 to 200,000 years ago [14].

The CCF can also be represented as a coalescent intensity function (CIF; see S4 Text), which measures the rate of change of common ancestry over time (Fig 5C, middle). The CIF reveals additional structure; for example, around 3,000 to 20,000 generations ago, those parts of the Siberian Eskimo's genome that have not yet coalesced with other genomes sampled from the same ancestry group have a very low CIF, while the CIF to the African ancestry samples (which have had very little coalescence until this point) is relatively high (though note the absolute rate remains very low over this period).

We can further summarize the coalescent profiles for the target individual by computing the maximum CIF over the sample. This statistic captures properties analogous to the effective population size parameter, $N_e$, in population genetics modeling, in which the expected coalescent intensity is inversely related to population size. We refer to the maximum CIF in the following as the $N_e$ equivalent (Fig 5C, bottom), though we note that (in the case in which there

are genuine populations) it is likely to be downward biased compared to existing methods; for example, we would expect PSMC [13] to infer absolute values of ancestral population size more accurately. We therefore use $N_e$ equivalents to provide a relative summary of genealogical histories across the entire cohort.

We estimated all pairwise CIFs in TGP (S2 Movie), which we aggregated across autosomes to generate a coalescent profile between each pair of individuals (see S6 Text). Likewise, we estimated the ancestry shared between each individual pair in SGDP (S3 Movie); CIFs were further aggregated among the 130 population groups (Fig 6). These reveal how the rates and structure of coalescence have changed over time, with the most recent epoch (up to approximately 200 generations ago) dominated by coalescence within populations, but also identify recent connections between groups such as between southern Siberian and northern East Asian populations (Fig 6A). Several populations such as the Kusunda (Nepal), Saami (Finland), and Negev Bedouins (Israel) show strong within-group coalescence up to this point, though by 500 generations ago they are coalescing primarily with other populations. The epoch around 800 generations ago is dominated by structure broadly corresponding to the continental level (Fig 6B), though some African populations—for example, the Mbuti (Congo), but more dramatically the Khomani San (South Africa) and Ju'hoansi (Namibia)—remain isolated up to approximately 1,500 generations ago (30,000 to 45,000 years), which overlaps with previous findings [63], and we see these two populations to be strongly connected for an extended period further back in time (S3 Movie). Around 800 generations ago, there is very little remaining structure among West Eurasian populations, but many additional intercontinental connections are now identified. For example, we see a north-to-south gradient of decreasing coalescence between American populations and Siberian or East Asian populations. In particular, we identify strong coalescence of American ancestry individuals with Siberian Eskimos, Aleutian Islanders, and Tlingit people in a period between 500 and 1,000 generations ago and very little structure among American, Siberian, and East Asian populations as a whole further back than 1,000 generations ago (S3 Movie), which agrees with previous results regarding the human migration into the Americas, extended isolation, and subsequent dispersal across the continent [46]. By 4,000 generations ago, we see high levels of coalescence between non-African and African populations (Fig 6C) and essentially no structure in the epoch around 20,000 generations ago (Fig 6D). We note that the exact timings of demographic events (signified by periods of intense or reduced coalescence), while showing consistent patterns within and among population groups, may carry additional noise due to uncertainty in allele age estimates.

The maximum CIF profiles ($N_e$ equivalents; Fig 6E) highlight several features, including differences among modern ancestry groups in the intensity of coalescence within the last 1,000 generations (particularly intense for American and Oceanic populations); a major period of intense coalescence among all non-African ancestry individuals around 1,000 to 2,000 generations ago, following the migration of modern humans out of Africa [64]; a weaker, but still marked, increase in coalescent intensity for African ancestry samples around 2,000 generations ago; and an older reduction in coalescent intensity, peaking around 5,000 to 8,000 generations ago, potentially driven by ancient population structure within Africa and (for non-African populations) possible admixture with archaic lineages [65–67]. We find minor quantitative, but not qualitative, differences among chromosomes (S8 Fig). We note that the bottleneck and steep change in population size around 60,000 to 80,000 generations ago, although consistent across ancestry profiles inferred for all individuals in the sample, is likely to be an artifact possibly caused by decreased accuracy of age estimates in the distant past, along with a breakdown of assumptions regarding the statistical independence of very old mutations.

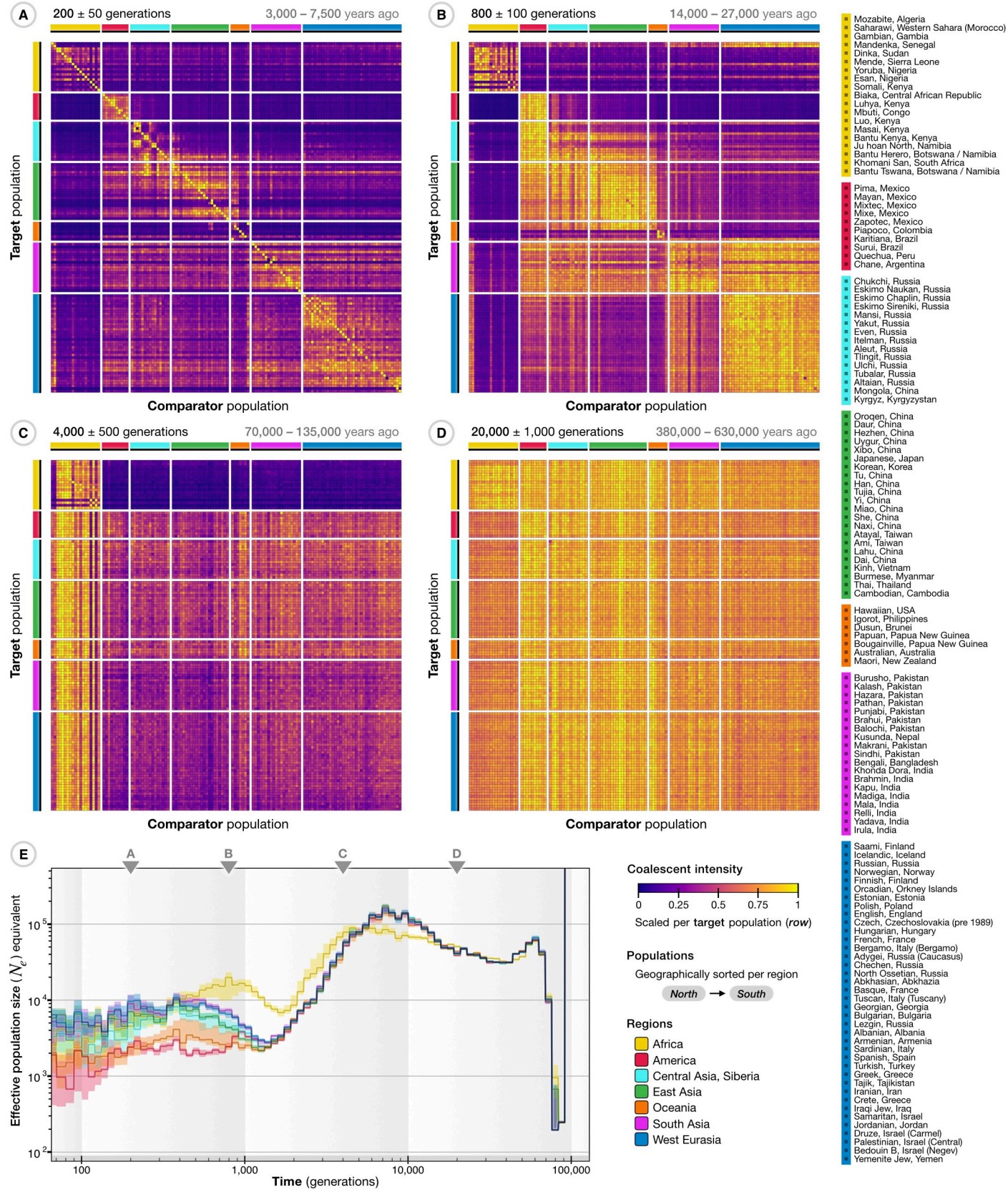

**Fig 6. Age-stratified connections between ancestry groups in the publicly available SGDP sample.** The CCF was inferred for all 556 haploid target genomes with all other comparator genomes in the SGDP sample and then aggregated by ancestry group (mean of CCFs from individuals within a population) and across chromosomes, with populations as defined in the SGDP (see legend on the right). (A–D) The ancestry shared between populations is indicated by the CIF over a given time interval (epoch), shown as a matrix with populations sorted from north to south within continental regions. Intensities were computed from aggregated CCFs to summarize relationships between populations; colors indicate intensity scaled per target population (rows) by the maximum over comparator populations. Ancestral connections are shown at different epochs back in time; around 200 generations ago (A), 800 generations (B), 4,000 generations (C), and 20,000 generations (D). The conversion (top right) assumes 20–30 years per generation. A more detailed summary, showing the ancestry shared between individuals, over a sliding time window (epoch) is shown in S3 Movie. (E) The maximum CIF for individuals from different ancestry groups (continental regions) expressed as effective population size ($N_e$) equivalents over time, estimated from CCFs aggregated per diploid individual and summarized by the median and interquartile range per group. Triangles indicate the epochs shown in panels A–D. A further breakdown of $N_e$ equivalents estimated from nonaggregated CCFs per chromosome is shown in S8 Fig. CCF, cumulative coalescent function; CIF, coalescent intensity function; SGDP, Simons Genome Diversity Project.

## Discussion

We have demonstrated how allele age estimates can provide insight to a range of problems in statistical and population genetics. However, there are several important assumptions and limitations of the approach. First, a key assumption is that of a single origin for each allele. Given the size of the human population and the mutation rate, it is likely that every allele has arisen multiple times over evolutionary history. Yet, unless the mutation rate is extremely high, it is still probable that most individuals carrying the allele do so through common ancestry. Moreover, multiple origins can potentially be identified through the presence of the allele on multiple haplotype backgrounds, as has, for example, been seen for the rs4988235 variant at *LCT* [37] (though we note that a previous study [68] concluded that the allele of variant rs4988235 was brought into African populations through historic gene flow, possibly through the Roman Empire), the O blood group [69], or alleles in the Human Leukocyte Antigene (*HLA*) region [70]. A variant lying in a region with high rates of noncrossover (gene conversion) may similarly be found on multiple haplotype backgrounds [71]. However, for genomes with very high mutation rates, such as HIV-1 [72], recurrence is sufficiently high to make estimates of allele age meaningless. In addition, while we have shown GEVA to be robust to realistic levels of sequencing and haplotype phasing error, the actual structures of error found in reference data sources, such as the TGP, have additional complexity whose effect is unknown [73].

Our approach also assumes a known and time-invariant rate of recombination. For most species, only indirect estimates of the per-generation recombination rate are available, and in humans [74] and mice [75], there is evidence for evolution in the fine-scale location of recombination hotspots through changes in the binding preferences of *PRDM9* (PR domain zinc finger protein 9). However, because broad-scale recombination rates evolve at a much lower rate than hotspot location [76] and because our approach for detecting recombination events is driven largely by the presence of recombinant haplotypes, we expect GEVA to be relatively robust for recent variants. Older variants may be more affected, but for such variants, most information comes from the mutation clock, which is likely to have been more stable over time.

We have shown that the ages of variants are highly correlated when estimated separately in independent samples, which agrees with the assumption that, for the majority of alleles, mutations occurred only once in the history of the population, such that the true age of an allele refers to a fixed point in time. In principle, the age can be estimated irrespective of the frequency distribution observed in different study cohorts. Our results show substantial heterogeneity in the relationship between frequency and age, revealing the impact of often unknown demographic variables on the distribution of alleles within and across different ancestry groups, which may further be confounded by the mode and strength of selection on particular alleles. The estimation profiles of variants shown in Fig 3 provide only a few examples of cases in which the frequency may not be reliable as a proxy for the age. Our method and the Atlas of

Variant Age may therefore provide more robust measures for analyses that otherwise rely on population or sample-specific allele frequencies.

The Atlas of Variant Age also has multiple applications beyond statistical and population genetics. For example, recent variants provide a natural index when searching for related samples in population-scale data sets. We have demonstrated how variant age information can be leveraged more comprehensively, using a nonparametric approach, to learn about genealogical history, relatedness between individuals, and ancestral connections between populations. Our approach to infer coalescent profiles (pairwise CCF and sample-wide CIF) captures ancestry proportions and coalescence rate variations as a function of time, from which we can distinguish recent from past demographic effects or estimate changes of relative relatedness over time. As illustrated for whole-genome sequencing data from the TGP and SGDP, our methodology accesses a time dimension to study relatedness between individual genomes and among entire sample cohorts. Applications to other data sets may use information from the Atlas of Variant Age without the need to re-estimate ages, thus demonstrating the value of this resource for exploratory data analysis.

Finally, as presented here, it is possible to combine information from multiple, potentially even distributed data sets by estimating coalescent time distributions for pairs of concordant and discordant haplotypes in each data resource separately or to update age estimates by the inclusion of additional samples. Estimation of the age of particular variants may gain additional accuracy by increasing the number of pairwise comparisons, depending on the history of the allele and the genealogical resolution attainable from the diversity of ancestral backgrounds inherent to available sample data. Future studies may reveal new variants, particularly within under-represented ancestry groups, that can also be added to the Atlas of Variant Age. The methods are also applicable to nonhuman species as long as estimates of mutation rate, recombination rate, and generation time are available. Future extensions to infer location of origin or the ancestral haplotype, integrating the growing wealth of genome data from ancient samples, will be an important step towards reconstructing the ancestral history of the entire species.

## Materials and methods

### Data sources

Estimation of allele age and shared ancestry was conducted on publicly available data sets: the TGP [2] and the SGDP [36]. We used phased haplotype data of chromosomes 1–22 from the final release TGP panel (Phase 3; GRCh37), available for 2,504 individuals from 26 populations worldwide (5 continental population groups). Additional data were available from the TGP for 31 related individuals, which we included in our shared ancestry analysis. We used phased haplotype data of chromosomes 1–22 from the publicly available SGDP panel (PS2; GRCh37), consisting of 278 individuals from 130 populations worldwide (7 continental population groups). Recombination rates were determined for each chromosome using the genetic maps available from the International HapMap Project (Phase 2; GRCh37) [78]. Genotype data from the Illumina Platinum Genomes Project (IPG) [79] (GRCh37; chromosomes 1–22) were used as a reference to measure genotype error in a matched subsample from the TGP. We used information from the Ensembl database (GRCh37; release 92, version 20180221) to determine the ancestral and derived allelic states for variants in both the TGP and SGDP panels, as predicted through multispecies alignments in the Ensembl EPO pipeline.

### Availability of results

The Atlas of Variant Age is publicly available as an online database: https://human.genome.dating/. It contains the age estimation results for more than 45 million variants in the human

genome, including the results of all pairwise analyses conducted at each focal site to estimate shared haplotype segments and infer TMRCA between haplotype pairs (more than 32 billion). We used an HMM to infer shared haplotype segments, for which we constructed an empirical error model (through comparison between genotypes from the IPG and TGP) to generate frequency-dependent emission and initial state probabilities; these are available online: https://github.com/pkalbers/geva/tree/master/hmm. Furthermore, we performed shared ancestry analyses for every pair of individual chromosomes in the TGP (including related individuals) and the SGDP, which are also available at https://human.genome.dating/.

### Source code availability

The source code of the GEVA method is available online at https://github.com/pkalbers/geva. The source code of the CCF dynamic programming algorithm is available at https://github.com/pkalbers/ccf. We modified the original source code of MSMC2 to optimize the performance of the PSMC algorithm in our simulation analysis, available at https://github.com/pkalbers/msmc2.

### Supporting information

**S1 Fig. Simple demographic model simulation.** Allele age estimation and TMRCA inference in a simulation of a 100-Mb region with sample size $N = 1,000$, effective population size $N_e = 10,000$, constant mutation rate ($\mu = 1 \times 10^{-8}$ per site per generation), and constant recombination rate ($r = 1 \times 10^{-8}$ per site per generation). (A) Relationship between the true allele age (geometric mean of lower and upper age of the branch on which a mutation occurred; $x$ axis) and estimated allele age ($y$ axis), estimated using the mutation clock, recombination clock, and joint clock models (left) and PSMC (right) for the same set of 5,000 variants, randomly sampled at allele count $1 < x < N$. Colors indicate the density scaled by the maximum per panel. Upper inserts indicate the fraction of sites where the point estimate (mode of the composite posterior distribution) of allele age lies above the upper age of the branch on which the mutation occurred (^), below the lower age (ˇ), or within the range of the branch (○). Lower inserts indicate the Spearman rank correlation statistic $\rho$, the square of the Pearson correlation coefficient (on log scale) $r^2$, the interval-adjusted error metric $\varepsilon$, and the RMSLE. Also shown is an LOESS fit (second-degree polynomials, neighborhood proportion $\alpha = 0.25$; dashed line). (B) Relationship between true TMRCA for a haplotype pair at a given site and corresponding inferred TMRCA (mean of posterior distribution), shown separately for concordant and discordant pairs. The same sets of haplotype pairs were analyzed under each clock model in GEVA (left) and PSMC (right). Colors indicate the density scaled by the maximum per panel. Lower inserts indicate the Spearman rank correlation statistic $\rho$, the square of the Pearson correlation coefficient (on log scale) $r^2$, and the RMSLE. GEVA, Genealogical Estimation of Variant Age; LOESS, locally estimated scatterplot smoothing; PSMC, pairwise sequentially Markovian coalescent; RMSLE, root mean-square $\log_{10}$ error; TMRCA, time to the most recent common ancestor.
(TIF)

**S2 Fig. Complex demographic model simulation without error.** Allele age estimation and TMRCA inference in a simulation that recapitulates the human expansion out of Africa [77], with $N = 5,000$, $N_e = 7,300$, constant mutation rate $\mu = 2.35 \times 10^{-8}$, and variable recombination rates from HapMap (Phase 2, GRCh37) [78] for chromosome 20 (63 Mb). Allele age was estimated for 5,000 variants sampled uniformly from the intersection of sites available at allele count $1 < x < N$ in data without error and after data were modified with error; see S3 Fig and

S4 Fig. Description of plots as in S1 Fig. GRCh37, Genome Reference Consortium Human Build 37; TMRCA, time to the most recent common ancestor.
(TIF)

**S3 Fig. Complex demographic model simulation with error.** Allele age estimation and TMRCA inference from simulated data in which haplotype data were modified with realistic error rates; calibrated from empirical estimates of genotype errors in TGP data [2], by comparison to corresponding genotype data from the IPG [79]. Allele age was estimated for the same set of 5,000 variants as analyzed in S2 Fig and S4 Fig. Description of plots as in S1 Fig. IPG, Illumina Platinum Genomes Project; TGP, 1000 Genomes Project; TMRCA, time to the most recent common ancestor.
(TIF)

**S4 Fig. Complex demographic model simulation with error and after phasing.** Allele age estimation from simulated data in which haplotype data were modified with realistic error rates, calibrated from empirical estimates of genotype errors in TGP data [2], by comparison to data from the IPG [79]. Haplotype data were additionally phased using SHAPEIT2 [80] after the introduction of data error. Allele age was estimated for the same set of 5,000 variants as analyzed in S2 Fig and S3 Fig. Description of plots as in S1 Fig. Note that the relationship between true and inferred TMRCA per haplotype pair cannot be ascertained conclusively after phasing of haplotype data. IPG, Illumina Platinum Genomes Project; TGP, 1000 Genomes Project; TMRCA, time to the most recent common ancestor.
(TIF)

**S5 Fig. Correlation between allele age estimated separately in TGP and SGDP data.** (A) The relationship between allele age using data from the TGP (*x* axis) and the SGDP (*y* axis), estimated from the mutation clock (left), recombination clock (center), and the joint clock model (right), for 13.7 million variants dated in both data sources. Colors indicate the density scaled by the maximum per panel. Lower inserts indicate the Spearman rank correlation statistic $\rho$ and the square of the Pearson correlation coefficient (calculated on log-scaled allele ages) $r^2$. (B) Differences in allele frequency of the variants compared (left); the histograms (right) show the frequencies as observed in the TGP (top) and the SGDP (bottom) for corresponding sets of variants. SGDP, Simons Genome Diversity Project; TGP, 1000 Genomes Project.
(TIF)

**S6 Fig. Allele age and frequency for variants shared between different human populations.** The relationship between allele age and frequency for variants dated in the TGP. Allele age was estimated under the joint clock model using TGP data (whole sample). Of the 43.2 million variants dated in the TGP (chromosomes 1–22), we excluded those with low estimation quality and inconsistent ancestral allele information, which retained 34.4 million variants. Allele frequencies were calculated within subsamples of AFR, AMR, EAS, EUR, and SAS ancestry groups, as defined in TGP sample data. Lines in each panel show the cumulative age distribution of variants within a given frequency bin (see legend), with frequencies as observed within the population group indicated at the top (columns); circles indicate median and interquartile range (25th, 50th, and 75th percentiles). (A) The subset of variants observed in only two population groups, referring to sites that have nonzero frequencies in either of the two populations considered and zero frequency in all other groups (white background color), and geographically restricted variants that are isolated within only a single population group, referring to sites that have nonzero frequencies in the population considered and zero frequency in all other groups (diagonal panels; gray background color). Panels that are diagonal opposites show results for the same set of variants but with frequencies as observed in the population

considered (by column). The number of variants retained is shown in each panel (bottom right). (B) The age distribution of strictly cosmopolitan variants (nonzero frequencies in every group) by frequency as seen within a population group. The distributions shown in each panel were obtained on same set of 3,634,716 variants. AFR, African; AMR, American; EAS, East Asian; EUR, European; SAS, South Asian; TGP, 1000 Genomes Project.
(TIF)

**S7 Fig. Allele age of potentially pathogenic variants.** Allele age was estimated (joint clock) using sample data from the TGP for variants annotated by the Ensembl VEP [81]. We excluded variants with low age estimation quality or inconsistent ancestral allelic states (determined through multispecies alignments; information available through Ensembl), which retained 64,432 (of 70,220) variants annotated by PolyPhen-2 [51] and 61,995 (of 67,539) variants annotated by SIFT [52]; of those, 61,615 (of 67,123) variants have been annotated by both methods. (A) The relationship between allele age and variant effects predicted by PolyPhen-2 (top), with effect categories given as benign, possibly damaging, probably damaging, and unknown; variant numbers per category are indicated in the legend. Each line shows the cumulative age distribution by effect category; circles indicate median and interquartile range. The frequency distribution of all variants considered is shown for the 5 major population groups defined in the TGP sample (middle), given as the number of variants within allele count bins (evenly spaced on linear scale but shown on log scale), with allele count as per population group in the TGP. The number of variants at nonzero frequencies in a population group is indicated in the legend. The relative proportion of variants across effect categories per allele count bin for the 5 major population groups in the TGP (bottom). (B) As for part (A), plots show the relationships between allele age and population frequency for variants predicted by SIFT, with effect categories given as tolerated and deleterious. (C) QQ-plots showing differences in allele age distributions for variants annotated by PolyPhen-2 (left) and SIFT (right), compared to a control set of variants (those annotated as benign by PolyPhen-2 or tolerated by SIFT), matched for allele frequency within a given population group. Matching was done by retaining only those variants observed at nonzero frequency within a population group and if variants of every effect category were represented at identical allele counts. The inset in each panel (bottom right) shows the number of variants retained per effect category. PolyPhen-2, "Polymorphism Phenotyping v2" software; SIFT, "Sorting Intolerant From Tolerant" software; TGP, 1000 Genomes Project; VEP, Variant Effect Predictor.
(TIF)

**S8 Fig. Effective population size ($N_e$ equivalent) over time for chromosomes 1–22 in the SGDP.** The CCF was inferred for each haploid target genome with all other comparator genomes in the SGDP sample, based on a total of 11.7 million variants dated (joint clock) on chromosomes 1–22, retained after excluding variants with low age estimation quality and inconsistent ancestral allele information. Coalescent intensity was computed per target genome and scaled by the maximum over the sample at a given time interval (epoch; evenly distributed on log scale). Each line shows the median and interquartile range of $N_e$ equivalents inferred for individuals in the different ancestry groups (continental regions; see legend). CCF, cumulative coalescent function; SGDP, Simons Genome Diversity Project.
(TIF)

**S1 Table. Summary of variants per chromosome in the Atlas of Variant Age.** The table shows the total number ($N_{all}$) of variants available in the Atlas of Variant Age on chromosomes 1–22, as well as the number of variants dated using data from the TGP alone ($N_{TGP}$) and the SGDP alone ($N_{SGDP}$). Additionally, variants present in both data sets were dated using

independently inferred pairwise TMRCA results from the TGP and SGDP to obtain a combined age estimate ($N_{\text{Combined}}$). The number of haplotype pairs at which shared haplotype segments and TMRCA were inferred is shown for the two data sources; numbers are shown as the sum of concordant and discordant pairs analyzed per chromosome. See S3 Text for details about the analysis of TGP and SGDP sample data. Full result data sets for each variant, including the results of each pairwise analysis and age estimates obtained under each clock model (mutation, recombination, and joint clock; see S1 Text), are publicly available online at https://human.genome.dating/. SGDP, Simons Genome Diversity Project; TGP, 1000 Genomes Project; TMRCA, time to the most recent common ancestor.
(PDF)

**S2 Table. Age and frequency of variants within population groups in TGP data.** We estimated allele age for variants identified in the TGP to characterize the age distribution of genetic variation across the human genome. Allele age was estimated under the joint clock model. Of the 43,232,520 variants dated in the TGP (chromosomes 1–22), we retained only those at quality score $QS > 0.5$ (see S1 Text) and at which the ancestral allele is known and mapped to the reference allele (see S3 Text), which retained 34,388,511 variants. The table shows the number of variants ($N$) and the median of allele age estimates ($Q_{50}$), as well as the 25th ($Q_{25}$) and 75th ($Q_{75}$) percentiles, per continental population group and stratified by allele frequency within that group. This is shown for (A) variants at nonzero frequencies within a given ancestry group, (B) geographically restricted variants that segregate only within a given group, and (C) strictly cosmopolitan variants that are shared among individuals from every continental group. AFR, African; AMR, American; EAS, East Asian; EUR, European; SAS, South Asian; TGP, 1000 Genomes Project.
(PDF)

**S1 Movie. Ancestry of individual HG00733.** The change in coalescence rate (intensity) over time for the two haploid genomes (chromosome 5) of target individual HG00733 (Puerto Rican in the American ancestry group) with all other genomes available through the TGP as comparator individuals; 5,070 haploid genomes in total, including related individuals that had been removed in the final release Phase 3 panel. Computations of the CCF and coalescent intensities are based on 2.3 million variants dated (joint clock model) on chromosome 5 from the TGP, retained after excluding variants of low estimation quality and inconsistent ancestral allele information. The movie shows the intensity scaled by the maximum over the TGP sample (top two panels), in which chromosomes are distinguished by maternal and paternal modes of inheritance (determined from coalescent intensity inferred with both parental genomes). The scaling factor (maximum of coalescent intensity across the sample) is shown separately (bottom left). An estimate of the effective population size ($N_e$ equivalent; bottom right) is obtained from the scaling factor and the number of generations within a given time interval (epoch). The two lines show the $N_e$ equivalent inferred for the maternally and paternally inherited chromosomes of the target individual. Epochs are shown on a sliding window, moving back in time, with epoch size as indicated (bottom). The full result data set for HG00733 is available online at https://human.genome.dating/ancestry/HG00733. CCF, cumulative coalescent function; TGP, 1000 Genomes Project.
(TXT)

**S2 Movie. Ancestry shared back in time for all individuals from the TGP.** The TGP sample consists of 2,535 individuals from 26 population groups, including related individuals that had been removed in the final release Phase 3 panel. We first computed the CCF for each of the 5,070 haploid genomes in turn with every other genome, separately for chromosomes 1–22.

This was based on age estimates of 34.4 million variants dated using TGP data (joint clock model), retained after excluding variants of low estimation quality and inconsistent ancestral allele information; CCFs were then aggregated per diploid individual and across chromosomes. We then calculated the CIF from the aggregated CCF of each pair of individuals. Ancestral relationships are shown in a matrix (left), where colors indicate the coalescent intensity between each pair of individuals, scaled per target individual (rows) by the maximum over the sample per time interval (epoch). Labels indicate the continental population as defined in the TGP, which can be further subdivided into smaller population groups (see legend at end of movie). The movie shows ancestry backwards in time, where coalescent intensity is calculated in a sliding window with epoch size as indicated (top right). An estimate of the effective population size ($N_e$ equivalent) is obtained per target individual from the maximum coalescent intensity across the sample and the time covered by a given epoch (bottom right); colors indicate an individual's ancestry group. Full result data sets for each individual are available online at https://human.genome.dating/ancestry/index#TGP. CCF, cumulative coalescent function; CIF, coalescent intensity function; TGP, 1000 Genomes Project.
(TXT)

**S3 Movie. Ancestry shared back in time for all individuals from the SGDP.** The SGDP sample consists of 278 individuals from 130 populations in the publicly available SGDP sample. We estimated the CIF from the aggregated CCF for each pair of individuals. CCFs were inferred separately per chromosome between each of the 556 haploid genomes as target in turn with all others as comparator genomes, based on 11.7 million variants dated under the joint clock model in the SGDP (chromosomes 1–22) and retained after excluding variants of low estimation quality and inconsistent ancestral allele information. These were then aggregated per diploid individual across chromosomes. As for S2 Movie, ancestral relationships are shown in a matrix (left), moving back in time, in which colors show the scaled coalescent intensity per epoch (as indicated on the timeline; top right). Individuals are labeled according to their population group as defined in the SGDP sample (see legend at end of movie). Full result data sets for each individual are available online at https://human.genome.dating/ancestry/index#SGDP. CCF, cumulative coalescent function; CIF, coalescent intensity function; SGDP, Simons Genome Diversity Project.
(TXT)

**S1 Text. Genealogical Estimation of Variant Age (GEVA).**
(PDF)

**S2 Text. Analysis of data error and application in simulations.**
(PDF)

**S3 Text. Estimation of variant age in publicly available data sets.**
(PDF)

**S4 Text. Shared ancestry inference.**
(PDF)

**S5 Text. Inference of shared ancestry in simulated sample data.**
(PDF)

**S6 Text. Inference of shared ancestry between individuals and population groups in publicly available data sets.**
(PDF)

## Acknowledgments

We thank members of the McVean group for comments and discussion. We thank Robert Esnouf (Oxford Big Data Institute) for computational support. Computation used the Oxford Biomedical Research Computing (BMRC) facility, a joint development between the Wellcome Centre for Human Genetics and the Big Data Institute supported by Health Data Research UK and the National Institute for Health Research (NIHR) Oxford Biomedical Research Centre. The views expressed are those of the authors and not necessarily those of the National Health Service (NHS), the NIHR, or the Department of Health.

## Author Contributions

**Conceptualization:** Patrick K. Albers, Gil McVean.

**Data curation:** Patrick K. Albers.

**Formal analysis:** Patrick K. Albers.

**Funding acquisition:** Gil McVean.

**Investigation:** Patrick K. Albers, Gil McVean.

**Methodology:** Patrick K. Albers, Gil McVean.

**Project administration:** Gil McVean.

**Resources:** Patrick K. Albers.

**Software:** Patrick K. Albers.

**Supervision:** Gil McVean.

**Validation:** Patrick K. Albers.

**Visualization:** Patrick K. Albers.

**Writing – original draft:** Patrick K. Albers, Gil McVean.

**Writing – review & editing:** Patrick K. Albers, Gil McVean.

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
