## [Editor Report · Decision Letter 0]

23 Jul 2019

Dear Dr Albers, 

Thank you for submitting your manuscript entitled "Dating genomic variants and shared ancestry in population-scale sequencing data" for consideration as a Methods and Resources paper by PLOS Biology.

Your manuscript has now been evaluated by the PLOS Biology editorial staff, as well as by an academic editor with relevant expertise, and I'm writing to let you know that we would like to send your submission out for external peer review.

**Important**: Please also see below for further information regarding completing the MDAR reporting checklist. The checklist can be accessed here: https://plos.io/MDARChecklist

Please re-submit your manuscript and the checklist, within two working days, i.e. by Jul 25 2019 11:59PM.

Kind regards,

Roli Roberts

Senior Editor

PLOS Biology

INFORMATION REGARDING THE REPORTING CHECKLIST:

PLOS Biology is pleased to support the "minimum reporting standards in the life sciences" initiative (https://osf.io/preprints/metaarxiv/9sm4x/). This effort brings together a number of leading journals and reproducibility experts to develop minimum expectations for reporting information about Materials (including data and code), Design, Analysis and Reporting (MDAR) in published papers. We believe broad alignment on these standards will be to the benefit of authors, reviewers, journals and the wider research community and will help drive better practise in publishing reproducible research. 

We are therefore participating in a community pilot involving a small number of life science journals to test the MDAR checklist. The checklist is intended to help authors, reviewers and editors adopt and implement the minimum reporting framework. 

IMPORTANT: We have chosen your manuscript to participate in this trial. The relevant documents can be located here:

MDAR reporting checklist (to be filled in by you): https://plos.io/MDARChecklist

**We strongly encourage you to complete the MDAR reporting checklist and return it to us with your full submission, as described above. We would also be very grateful if you could complete this author survey:

https://forms.gle/seEgCrDtM6GLKFGQA

Additional background information:

Interpreting the MDAR Framework: https://plos.io/MDARFramework

Please note that your completed checklist and survey will be shared with the minimum reporting standards working group. However, the working group will not be provided with access to the manuscript or any other confidential information including author identities, manuscript titles or abstracts. Feedback from this process will be used to consider next steps, which might include revisions to the content of the checklist. Data and materials from this initial trial will be publicly shared in September 2019. Data will only be provided in aggregate form and will not be parsed by individual article or by journal, so as to respect the confidentiality of responses. 

Please treat the checklist and elaboration as confidential as public release is planned for September 2019.

We would be grateful for any feedback you may have.

---

## [Decision Letter · Decision Letter 1]

12 Sep 2019

Dear Dr Albers,

Thank you very much for submitting your manuscript "Dating genomic variants and shared ancestry in population-scale sequencing data" for consideration as a Methods and Resources paper at PLOS Biology. Your manuscript has been evaluated by the PLOS Biology editors, an Academic Editor with relevant expertise, and by three independent reviewers.

You'll see that the reviewers are broadly positive about your study, but have a number of requests for clarifications, additional analyses and presentational improvements. The Academic Editor asked us to emphasise that s/he shares reviewer #2's uneasiness about your reliance on a single modal estimate (which is how most people will use the database), and thinks that this could be quite misleading in downstream analysis, since it discards information on the range of plausible estimates; s/he requests that you at least include some explicit discussion of this issue.

In light of the reviews (below), we are pleased to offer you the opportunity to address the comments from the reviewers in a revised version that we anticipate should not take you very long. We will then assess your revised manuscript and your response to the reviewers' comments and we may consult the reviewers again. IMPORTANT: Note that reviewer #2's comments are available in the downloadable attachment.

Your revisions should address the specific points made by each reviewer. Please submit a file detailing your responses to the editorial requests and a point-by-point response to all of the reviewers' comments that indicates the changes you have made to the manuscript. In addition to a clean copy of the manuscript, please upload a 'track-changes' version of your manuscript that specifies the edits made. This should be uploaded as a "Related" file type. You should also cite any additional relevant literature that has been published since the original submission and mention any additional citations in your response. 

Before you revise your manuscript, please review the following PLOS policy and formatting requirements checklist PDF: http://journals.plos.org/plosbiology/s/file?id=9411/plos-biology-formatting-checklist.pdf. It is helpful if you format your revision according to our requirements - should your paper subsequently be accepted, this will save time at the acceptance stage.

Please note that as a condition of publication PLOS' data policy (http://journals.plos.org/plosbiology/s/data-availability) requires that you make available all data used to draw the conclusions arrived at in your manuscript. If you have not already done so, you must include any data used in your manuscript either in appropriate repositories, within the body of the manuscript, or as supporting information (N.B. this includes any numerical values that were used to generate graphs, histograms etc.). For an example see here: http://www.plosbiology.org/article/info%3Adoi%2F10.1371%2Fjournal.pbio.1001908#s5.

For manuscripts submitted on or after 1st July 2019, we require the original, uncropped and minimally adjusted images supporting all blot and gel results reported in an article's figures or Supporting Information files. We will require these files before a manuscript can be accepted so please prepare them now, if you have not already uploaded them. Please carefully read our guidelines for how to prepare and upload this data: https://journals.plos.org/plosbiology/s/figures#loc-blot-and-gel-reporting-requirements.

Upon resubmission, the editors assess your revision and assuming the editors and Academic Editor feel that the revised manuscript remains appropriate for the journal, we may send the manuscript for re-review. We aim to consult the same Academic Editor and reviewers for revised manuscripts but may consult others if needed.

We expect to receive your revised manuscript within one month. Please email us (plosbiology@plos.org) to discuss this if you have any questions or concerns, or would like to request an extension. At this stage, your manuscript remains formally under active consideration at our journal; please notify us by email if you do not wish to submit a revision and instead wish to pursue publication elsewhere, so that we may end consideration of the manuscript at PLOS Biology.

When you are ready to submit a revised version of your manuscript, please go to https://www.editorialmanager.com/pbiology/ and log in as an Author. Click the link labelled 'Submissions Needing Revision' where you will find your submission record. 

Sincerely,

Roli Roberts

Senior Editor

PLOS Biology

REVIEWERS' COMMENTS:

Reviewer #1:

This paper describes a very useful and carefully built resource: an atlas of allele age estimates for the variants in a large database of publicly available genomes. The algorithm used to infer these allele ages is clearly described in the supplement and represents a cogent heuristic for tackling this difficult problem. GEVA has the potential to be a 1-stop shop for local ancestry inference and demographic inference, two important classes of methods that usually involve different assumptions and software. 

I have no significant concerns about the technical soundness of this manuscript. I do think it could be cleaned up a bit to improve its “user-friendliness”—the supplement is a very rich source of information, but it is not very well organized or indexed. I think the presentation of supplementary figures before the supplement index is counterproductive, and would prefer the index to be on the first page of this file so readers can more easily refer back to it. 

The discussion of the paper is very short, especially given the breadth of results being presented, and it could be usefully expanded to flesh out how GEVA fits into the existing landscape of similar methods. For example, GEVA does some of the same things as tsinfer, another method from the McVean group, including inferring local ancestry profiles. Can the authors recommend when one local ancestry method may be more appropriate than the other, and how they are likely to compare to methods like chromopainter, RFmix, etc? In addition, the paper includes many PSMC-like plots, but it isn’t clear whether the authors would recommend their method as an alternative to users who may be interested in running PSMC on new genomes from humans or other organisms. A biorxiv preprint by Leo Speidel, Simon Myers, et al. also estimates allele ages, and it would be useful to say something about the differences between the methods in performance and applicability. 

The GEVA source code is available, but may or may not be as user-friendly as the database of variant ages that the authors obtained from a fixed set of publicly available genomes. Are the authors planning to keep updating the database as more genomes are sequenced, or are readers encouraged to run the method on other datasets themselves? Is it recommended for use on data from organisms other than humans? Do the authors expect that inferring demography directly from their coalescence time density functions might be more or less accurate than inferring demography using PSMC or other methods that incorporate site frequency spectrum information?

Minor comments:

-A PLoS Genetics paper recently published by Alexander Platt, Jody Hey, et al. also estimates allele ages. This should probably be referenced.

-The supplement presents a nice empirically calibrated error model that is trained on the discrepancies between the Illumina Platinum Genomes and less accurate sequences generated from the same cell lines. The model appears to generate posterior probability estimates as to whether a given variant is an error or not. These error probabilities are likely to be useful in downstream applications of the variant allele age database. Are they searchable within the database? When a variant is queried, will it be flagged as probably an error if appropriate?

Reviewer #2:

IMPORTANT! Please see attached PDF

Reviewer #3:

[identifies himself as Aylwyn Scally]

Just some quick comments from me. I like this paper, found it very interesting, and look forward to seeing it published. No major issues to raise, but I do have a couple of observations regarding the emphasis of the text which I think the authors should consider addressing. 

Firstly, it seems a bit odd that the first section of the paper, and indeed the first figure, places such emphasis on the comparison with using PSMC to estimate allele ages. This feels like inside baseball -- interesting from the perspective of algorithm development but not so important for understanding how it works. Perhaps I am missing something, but there didn't seem to be anything particularly surprising; using PSMC the method performed similarly, as one would expect, barring discretisation, in terms of accuracy if not speed, and the take-away message is that the authors' tailored approach was faster without adding bias or error. This is good but I think the simulation comparison alone without PSMC would be more meaningful for most readers, in terms of convincing the reader that the method does a good job at recovering allele age, which I think is the main task at this point in the paper. I'd put the PSMC aspect in the supplement. 

Secondly, and perhaps more importantly, I'd like to see the paper give a bit more focus to the comparison with allele frequencies. A lot of people will approach this study and the resources presented as a more accurate way to incorporate allele age into their analyses than simply using allele frequency as a proxy. Figures 4A and S6 give some useful information, but I think the presentation and discussion is a bit limited. One thing to note, for example, is how contingent allele age is on population of ascertainment. For example the rarest alleles (<1%) if ascertained in Africa are on average about twice the age of equivalent alleles ascertained in Europe or East Asia. More pertinently, are there any cases where using frequency as a proxy might be misleading or biased, relative to using the ages estimated here? My overall impression is that frequency is a pretty good proxy for age if ascertained straightforwardly in a single G1k population - but the mapping is nonlinear and does not transfer between populations. 

There are also some interesting signals in figure S6. The text points out the striking signal of recent African admixture in America, but there is also an apparent sign of recent gene flow from Africa into Europe, manifesting as an excess of old but very rare alleles there. At least, that is how I read the plot in the fourth panel of the top row -- do the authors concur? A consequence of this is presumably also seen in figure 4A, where European variants older than ~25000 years are apparently more likely to be in the rarest frequency bin than the next most common. 

Aylwyn Scally

---

## [Decision Letter · Decision Letter 2]

21 Nov 2019

Dear Dr Albers,

Thank you for submitting your revised Methods and Resources entitled "Dating genomic variants and shared ancestry in population-scale sequencing data" for publication in PLOS Biology. I have now obtained advice from one of the original reviewers and have discussed their comments with the Academic Editor. 

We're delighted to let you know that we're now editorially satisfied with your manuscript. However before we can formally accept your paper and consider it "in press", we also need to ensure that your article conforms to our guidelines. A member of our team will be in touch shortly with a set of requests. As we can't proceed until these requirements are met, your swift response will help prevent delays to publication. Please also make sure to address the Data Policy-related requests noted at the end of this email.

*Copyediting*

*Published Peer Review History*

*Early Version*

*Submitting Your Revision*

Sincerely,

Roli Roberts

Senior Editor

PLOS Biology

DATA POLICY:

Regardless of the method selected, please ensure that you provide the individual numerical values that underlie the summary data displayed in the figure panels as they are essential for readers to assess your analysis and to reproduce it. I note that most of the data presented are probably available in https://human.genome.dating - if so, please could you cite this in all relevant main and Supp Fig legends (I know you already do in some). However, there are several Figures where I'm uncertain whether the data come directly from https://human.genome.dating (e.g. Fig 2); if not, we will need you to provide them in another form (see above) and cite the location accordingly.

REVIEWERS' COMMENTS:

Reviewer #2:

[identifies himself as Yuval Simons]

I thank the authors for taking the time to address all the issues raised in the previous round of reviews in such meticulous detail.

I think this elegant and exciting work is ready for publication.

---

## [Editor Report · Decision Letter 3]

2 Jan 2020

Dear Dr Albers,

On behalf of my colleagues and the Academic Editor, Nick H. Barton, I am pleased to inform you that we will be delighted to publish your Methods and Resources in PLOS Biology. 

Early Version

PRESS 

Kind regards,

Hannah Harwood

Publication Assistant, 

PLOS Biology

on behalf of

Roland Roberts,

Senior Editor

PLOS Biology